# D-index and invasive fungal infections (IFIs) in adult acute myeloid leukemia (AML) patients with the first episode of febrile neutropenia

**Thanawat Rattanathammethee**[1]*, **Kawin Munsamai**[2], **Teerachat Punnachet**[1], **Nonthakorn Hantrakun**[1], **Pokpong Piriyakhuntorn**[1], **Sasinee Hantrakool**[1], **Chatree Chai-Adisaksopha**[1], **Ekarat Rattarittamrong**[1], **Adisak Tantiworawit**[1], **Lalita Norasetthada**[1]

1 Division of Hematology, Department of Internal Medicine, Faculty of Medicine, Chiang Mai University, Chiang Mai, Thailand, 2 Department of Internal Medicine, Faculty of Medicine, Chiang Mai University, Chiang Mai, Thailand

* thanawat.r@cmu.ac.th

## Abstract

### Introduction

This study aimed to evaluate the performance of the D-index, a calculated measure of neutropenic burden, in predicting the risk of invasive fungal infections (IFIs) in acute myeloid leukemia (AML) patients.

### Methods

A retrospective study of adult AML patients who received the first induction chemotherapy and developed febrile neutropenia was conducted. Clinical characteristics, laboratory data, and the calculation of the D-index and cumulative D-index (c-D-index) were collected and analyzed between patients with and without IFIs.

### Results

A total of 101 patients were included, with 16 (15.8%) patients who developed IFIs. Clinical characteristics, antifungal prophylaxis, and AML cytogenetic risk were similar between patients with or without IFIs. The results showed that the D-index and c-D-index were more effective in predicting IFIs than the duration of neutropenia. With the D-index cutoff of 7083, the sensitivity, specificity, positive predictive value (PPV), and negative predictive value (NPV) were 81.3%, 83.5%, 48.2%, and 95.9%, respectively. c-D-index at 5625 revealed sensitivity, specificity, PPV, and NPV for IFIs of 68.8%, 68.2%, 28.9%, and 92.1%, respectively. Using this cutoff of c-D-index, patients without IFIs were overtreated with an antifungal regimen in 45 (52.9%) cases.

**Data Availability Statement:** All relevant data are within the paper.

**Funding:** The author(s) received no specific funding for this work.

**Competing interests:** The authors have declared that no competing interests exist.

## Conclusion

The D-index and c-D-index were helpful indicators for defining the risk of IFIs in AML patients with febrile neutropenia.

## Introduction

Acute myeloid leukemia (AML) is the most common adult acute leukemia with a curative rate from standard high-intensity chemotherapy around 60–70% [1]. Unfortunately, severe neutropenia usually occurs during intensive chemotherapy with the curative aim [2, 3] and AML is the most common type of acute leukemia developing febrile neutropenia [4]. Neutropenia is a major risk factor for invasive fungal infections (IFIs), and IFIs are also the major life-threatening infectious complications among patients with febrile neutropenia following high-intensity chemotherapy or hypomethylating agents plus venetoclax for AML treatment [5, 6].

The severity of neutropenia was assessed from the duration of grade 4 neutropenia, defined as an absolute neutrophil count (ANC) <500 /μl and the duration of profound neutropenia (ANC <100 /μl) [7] associated with the neutropenic-related infectious complications. However, the only duration without using the serially dynamic change of ANC during bone marrow suppression may not reflect the overall neutropenic burden in patients. For this reason, the D-index was developed by the calculated area over the neutrophil curve from the plot of ANC <500 /μl and duration of grade 4 neutropenia [8]. Cumulative D-index (c-D-index) was also defined from the grade 4 neutropenia to the first manifestation of IFIs or onset of febrile neutropenia to determine the utility of D-index in predicting infectious-related complications [8–11]. Both D-index and c-D-index showed high sensitivity and negative predictive value for IFIs in AML patients, with the incidence of IFIs ranging from 0.5 to 15% [8, 9, 12]. Moreover, c-D-index-guided treatment significantly reduced the proportion of antifungal therapy usage without survival difference compared to the empirical strategy [10].

However, there was a limited study of the D-index in resource-limited countries with a high incidence of IFIs, especially in AML patients [13]. Moreover, there was the difference between AML treatment, supportive care, and use of antifungal prophylaxis among different centers. Therefore, this study aimed to investigate the impacts of D-index performance on IFIs and other infectious complications in adult AML patients who developed the first episode of febrile neutropenia (FN) after receiving high-intensity chemotherapy in Thailand.

## Materials and methods

### Patient selection and definitions

Data of AML patients between 1 January 2014 to 31 December 2020 were collected and a retrospective cohort was conducted between 23 June 2021 to 22 June 2022 at Chiang Mai University Hospital, Thailand. All clinical characteristics were collected by the retrospective chart review with de-identified patients' data, potentially bias from missing data. This study was conducted following the Declaration of Helsinki and the International Conference on Harmonization guidelines for Good Clinical Practice. The institutional ethical review board of the Faculty of Medicine, Chiang Mai University, Thailand, approved the study (study code: MED-2564-08127). Inclusion criteria included newly diagnosed de novo AML patients aged 18 years or above who received the first cycle of standard induction chemotherapy (7+3 regimen) with an adequate serial record of complete blood count (CBC) during admission. The 7+3 regimen consisted of a seven-day cytarabine 100 mg/m$^2$ intravenous continuous infusion over 24 hours combined with a three-day idarubicin 12 mg/m$^2$ bolus intravenously. Cytogenetics and genetic

abnormalities of AML were classified as favorable, intermediate, and adverse according to the European LeukemiaNet (ELN) [14]. Patients who received palliative care or low-intensity treatment, refractory to induction chemotherapy, lacking available CBC data, ANC was not recovered over 500 /μl after treatment, developed IFIs before initiating the 7+3 regimen, and patients diagnosed with acute promyelocytic leukemia (APL) were excluded. To determine the utility of D-index in patients with febrile neutropenia (FN), only AML patients who developed FN were included, defined as a single oral temperature of $\geq$ 101ºF (38.3ºC) or a temperature of $\geq$ 100.4ºF (38ºC) sustained over 1 hour with grade 4 neutropenia (ANC less than 500 /μl) [15]. Fluoroquinolone prophylaxis was not routinely used in the practice of our center. All patients received antifungal prophylaxis with itraconazole 200 mg twice daily and acyclovir 400 mg twice daily for prophylactic of herpes viral infection or reactivation. AML patients who developed IFIs were classified as possible, probable, or proven according to the revised criteria of invasive fungal disease from the European Organization for Research and Treatment of Cancer and the Mycoses Study Group Education and Research Consortium (EORTC/MSG) [16]. Our institutional policy regarding the antifungal treatment included the empirical strategy of amphotericin B 1 mg/kg/day in patients who developed FN without clinical response after 48–72 hours of broad-spectrum antibiotics administration or patients who developed the suggestive clinical features of fungal infection such as pleuritic chest pain, hemoptysis, or skin nodules. In terms of diagnostic-driven strategy, patients with a positive result of serum galacto-mannan or highly suggestive fungal infection from computed tomography (CT) imaging or biopsy-confirmed fungal infection were provided with voriconazole 6 mg/kg twice daily for the first 24 hours, then 4 mg/kg twice a day for treatment of invasive pulmonary aspergillosis (IPA) and/or Fusariosis. All AML patients in this study were hospitalized under the laminar flow after starting the 7+3 regimen until ANC recovery.

## D-index and cumulative D-index (c-D-index) calculation

The indices investigated in this study were schematically illustrated (Fig 1). The D-index was calculated from the plot between the serial record of ANC on the Y-axis and the duration in days on the X-axis. D-index was derived from the area over the neutrophil curve during grade 4 neutropenia, calculated from the difference between an observed area under the curve (Ao) and the expected neutrophil area if the patient did not develop grade 4 neutropenia (Ae). Ao was calculated by the trapezoidal method, while Ae is the product of 500/μl of ANC and the days of grade 4 neutropenia (Ae: 500 /μl x days of grade 4 neutropenia). The cumulative D-index (c-D-index) was defined as an accumulation of neutropenia until IFIs occurrence, which was calculated from grade 4 neutropenia to the first date of clinical manifestation of IFIs. The clinical manifestations of IFIs were categorized into oral cavity lesions (oral thrush), respiratory symptoms (cough, hemoptysis, pleuritic chest pain, nasal discharge), and skin changes (rash, nodules, swellings, painful skin including catheter site insertion). All clinical manifestations were confirmed according to clinical symptoms by the aseptic collection of related clinical specimens, diagnostic imaging, and serum biomarkers of IFIs such as serum galactomannan. If the confirmation test showed no evidence of IFIs, the c-D-index was equal to D-index. The calculation of D-index and c-D-index were facilitated by using the spreadsheet (S1 File).

## Sample size and statistical analysis

The performance of D-index to determine IFIs was considered the primary outcome of this study. According to an original study of D-index predicted IFIs in various settings of AML patients with neutropenia [8], the D-index cutoff of 6,200 showed 100% sensitivity and 58%

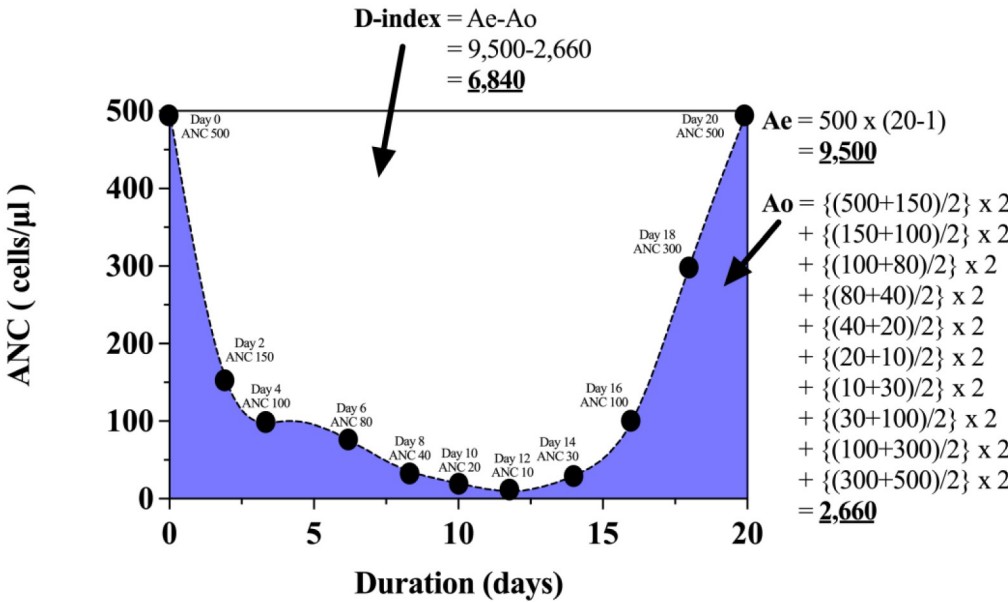

**Fig 1. Illustrated figure of an example of D-index calculation.** D-index was derived from the difference between the expected neutrophil area (Ae) and the observed area under the curve (Ao). Ae was calculated from the product of 500/μl of ANC and the days of grade 4 neutropenia. Ao was calculated from the sum of the trapezoidal area under the neutrophil curve.

specificity along with the 15% prevalence of IFIs in AML patients in our center [17]. The sample size for adequate sensitivity was 100 patients with an acceptable margin of error of 5%. The descriptive statistics were described as mean ± standard deviation (SD), median with interquartile range (IQR), or percentage values as appropriate. Dichotomous variables were compared using the Chi-square or Fisher's exact test. The test of normality was assessed by the Shapiro-Wilk test. Comparison of continuous variables between IFIs and no-IFIs group was analyzed by Student's *t*-test (for parametric analysis) or Mann-Whitney U test (for non-parametric analysis). A receiver operating characteristics (ROC) curve analysis was performed to determine the optimal cutoff value of the D-index and c-D-index to predict the IFIs [18]. Sensitivity, specificity, positive predictive value (PPV), and negative predictive value (NPV) were calculated using the cutoff values obtained from the ROC analysis. All patients with missing data were excluded. The data were entered into a custom database (Excel, Microsoft Corp) and analyzed using Prism 8 software (GraphPad, Inc., La Jolla, CA) and Stata/SE version 14.1 for Mac (Stata Corp, TX). A *P*-value of less than 0.05 was considered statistically significant.

## Results

A total of 237 AML patients were recruited, 136 patients were excluded, and 101 AML patients who developed the first episode of febrile neutropenia after induction chemotherapy were subsequently enrolled, including 16 patients with IFIs and 85 AML patients without IFIs (Fig 2). The enrolled patients' characteristics were summarized (Table 1). There were no significant differences in age, body mass index (BMI), serum albumin, the proportion of patients on sex, comorbidities, and cytogenetic risk of AML between the IFIs and no-IFIs groups. The granulo-cyte-colony stimulating factor (G-SCF) was used in patients with IFI more than patients without IFI (87.5% vs. 37.7%). Lung was the most common site of IFIs, accounting for 14 (87.5%) patients. All cases with pulmonary IFIs were invasive pulmonary aspergillosis (IPA), including

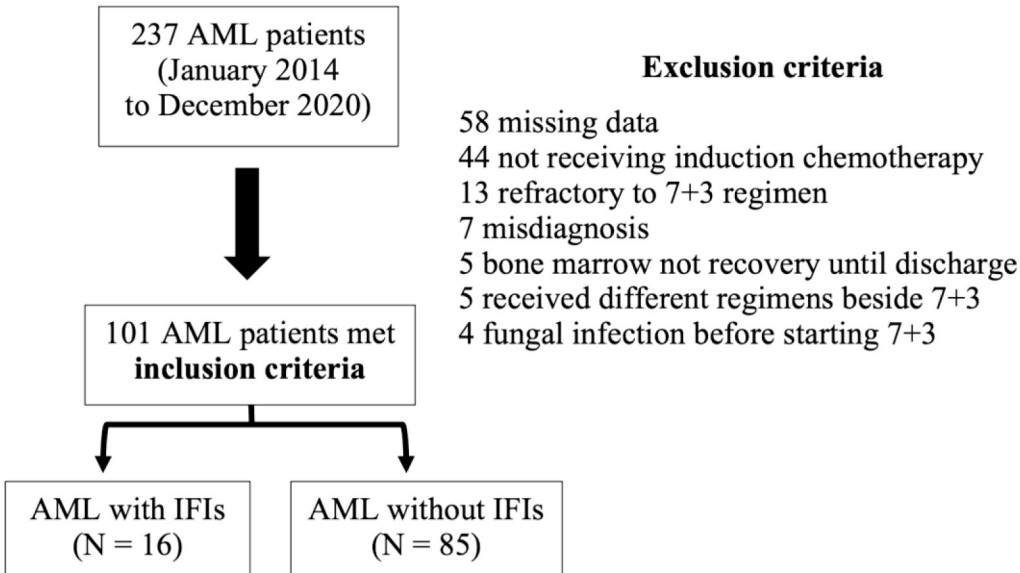

**Fig 2. Flow diagram of total recruitment alongside inclusion and exclusion criteria.**

13 possible IPA and one probable IPA. The other 2 cases of IFIs were localized cutaneous fusariosis. All IFI cases were initially treated with 1 mg/kg/day of intravenous amphotericin B with a median duration of 5.2 days (IQR: 2.1–7.4), then switched to voriconazole after obtained the positive result of serum galactomannan or highly suggestive fungal infection from computed tomography (CT) imaging or biopsy-confirmed fungal infection. Other infections showed no difference between IFIs and no-IFIs groups. Overall, the mean follow-up until ANC recovery was 22.6 ± 10.5 days. The mean duration of grade 4 neutropenia and profound neutropenia (ANC < 100 /μl) of patients with IFI were significantly longer compared with patients without IFIs (25.0 ± 12.1 days vs. 20.5 ± 5.3 days ($P$ = 0.015) and 18.9 ± 7.9 days vs. 12.7 ±5.6 days ($P$ < 0.001), respectively), as was the median D-index (8,803.3 (IQR: 7,746–9,923.8) vs. 4,312 (IQR: 2,311–6,274); $P$ < 0.001). In addition, the median c-D-index of the IFI group was 6,535.5 (IQR: 5,498.5–8,944.5) compared with a total D-index of the no-IFI group of 4,312 (IQR: 2,311–6,274; $P$ < 0.001).

ROC curve analysis of the D-index, with an area under the ROC curve (AuROC) of 0.937, showed better sensitivity and specificity to predict IFIs compared to the duration of grade 4 neutropenia with AuROC of 0.628 ($P$ < 0.001) and the duration of profound neutropenia with AuROC of 0.744 ($P$ < 0.001) (Fig 3A and 3B). The c-D-index ROC curve analysis showed AuROC of 0.802 with better performance than grade 4 neutropenia duration ($P$ = 0.012). Still, there was no statistical difference with profound neutropenia ($P$ = 0.295) (Fig 3C and 3D).

With the 15.8% prevalence of IFIs in this study, an optimal cutoff of 7,083 for the D-index in predicting IFIs showed the sensitivity, specificity, PPV, and NPV of 81.3%, 83.5%, 48.2%, and 95.9%, respectively. Below this cutoff, the IFIs were missed in 3 (18.8%) patients and patients without IFIs were overtreated with an antifungal regimen in 54 (63.5%) cases. Patients without IFIs above this cutoff missed detection in 7 (8.2%) patients. Regarding the cutoff point of 5,625 for c-D-index, the sensitivity, specificity, PPV, and NPV were 68.8%, 68.2%, 28.9%, and 92.1%, respectively. Using this cutoff of c-D-index, IFIs were missed in 5 (31.3%) patients and patients without IFIs were overtreated with an antifungal regimen in 45 (52.9%) cases. Patients without IFIs above this cutoff missed detection in 11 (12.9%) patients.

**Table 1. Demographics data and clinical characteristics between AML patients with IFIs and without IFIs.**

| | AML with IFIs (n = 16) | AML without IFIs (n = 85) | *P*-value |
|---|---|---|---|
| Age, years (mean ± SD) | 41.6 ± 13.4 | 39.7 ± 12.6 | 0.585 |
| Female, n (%) | 9 (56.3) | 46 (54.1) | 0.880 |
| BMI, kg/m$^2$ (mean ± SD) | 21.9 ± 4.2 | 21.8 ± 3.4 | 0.977 |
| Serum albumin, g/dL (mean ± SD) | 3.7 ± 0.7 | 3.8 ± 0.6 | 0.645 |
| Comorbidities*, n (%) | 2 (12.5) | 17 (20.0) | 0.730 |
| Cytogenetic risk of AML, n (%) | | | 0.276 |
| Favorable | 0 | 10 (11.8) | |
| Intermediated | 14 (87.5) | 58 (68.2) | |
| Adverse | 2 (12.5) | 17 (20.0) | |
| G-CSF used, n (%) | 14 (87.5) | 32 (37.7) | < 0.001 |
| Type of IFIs, n (%) | | | - |
| Pulmonary | 14 (87.5) | - | |
| Extrapulmonary | 2 (12.5) | - | |
| Other infections, n (%) | | | |
| Bacteremia | 0 | 4 (4.7) | 0.989 |
| Respiratory tract infection | 2 (12.5) | 9 (10.6) | 0.685 |
| Skin and soft tissue infection | 5 (31.3) | 40 (47.1) | 0.243 |
| Gastrointestinal infection | 5 (31.3) | 21 (24.7) | 0.583 |
| Urinary tract infection | 0 | 7 (8.2) | 0.593 |
| WBC counts per week (times) median (IQR) | 3 (2–5) | 3 (2–6) | 0.573 |
| Days of grade 4 neutropenia (< 500 /µl) mean ± SD | 25.0 ± 12.1 | 20.5 ± 5.3 | 0.015 |
| Days of profound neutropenia (< 100 /µl) mean ± SD | 18.9 ± 7.9 | 12.7 ±5.6 | < 0.001 |
| D-index (days • neutrophils/ µl) median (IQR) | 8,803.3 (7,746–9,923.8) | 4,312 (2,311–6,274) | < 0.001 |
| c-D-index (days • neutrophils/ µl) median (IQR) | 6,535.5 (5,498.5–8,944.5) | 4,312 (2,311–6,274) | < 0.001 |

**AML:** acute myeloid leukemia, **IFIs:** invasive fungal infections, **BMI:** body mass index,

**G-CSF**: granulocyte-colony stimulating factor, **WBC:** white blood cells

*Comorbidities: AML with IFIs (n = 2) included each one of hypertension and hyperlipidemia;

AML without IFIs (n = 17) included hypertension (n = 6), diabetes mellitus (n = 3), HBV carrier (n = 6),

HCV infection in sustained virological response (n = 1), Peutz-Jeghers syndrome (n = 1)

## Discussion

This study demonstrated the performance of the D-index and c-D-index for evaluating neutropenic burden-associated IFIs in AML patients with the first episode of febrile neutropenia. AML patients with IFIs had a longer duration of grade 4 neutropenia and profound neutropenia than those without IFIs. Using the D-index, which added on the serially dynamic change of ANC during neutropenia with the duration of neutropenia, also showed significantly higher in AML patients with IFIs. Furthermore, the accumulation of neutropenic burden until the first manifestation of IFIs, known as c-D-index, similarly revealed a greater extent in the IFIs group. Interestingly, these neutropenic burden scores also outperformed the duration of grade 4 neutropenia and profound neutropenia for predicting IFIs, except for the c-D-index and profound neutropenia duration. The prevalence of IFIs in AML patients with first-time febrile neutropenia in this study was 15.8%. The NPV of the D-index and c-D-index was 95.9% and 92.1% using cutoff values of 7,083 and 5,625, respectively. The high NPV of these indices could potentially exploit to exclude IFIs if the patients had the D-index and c-D-index below the

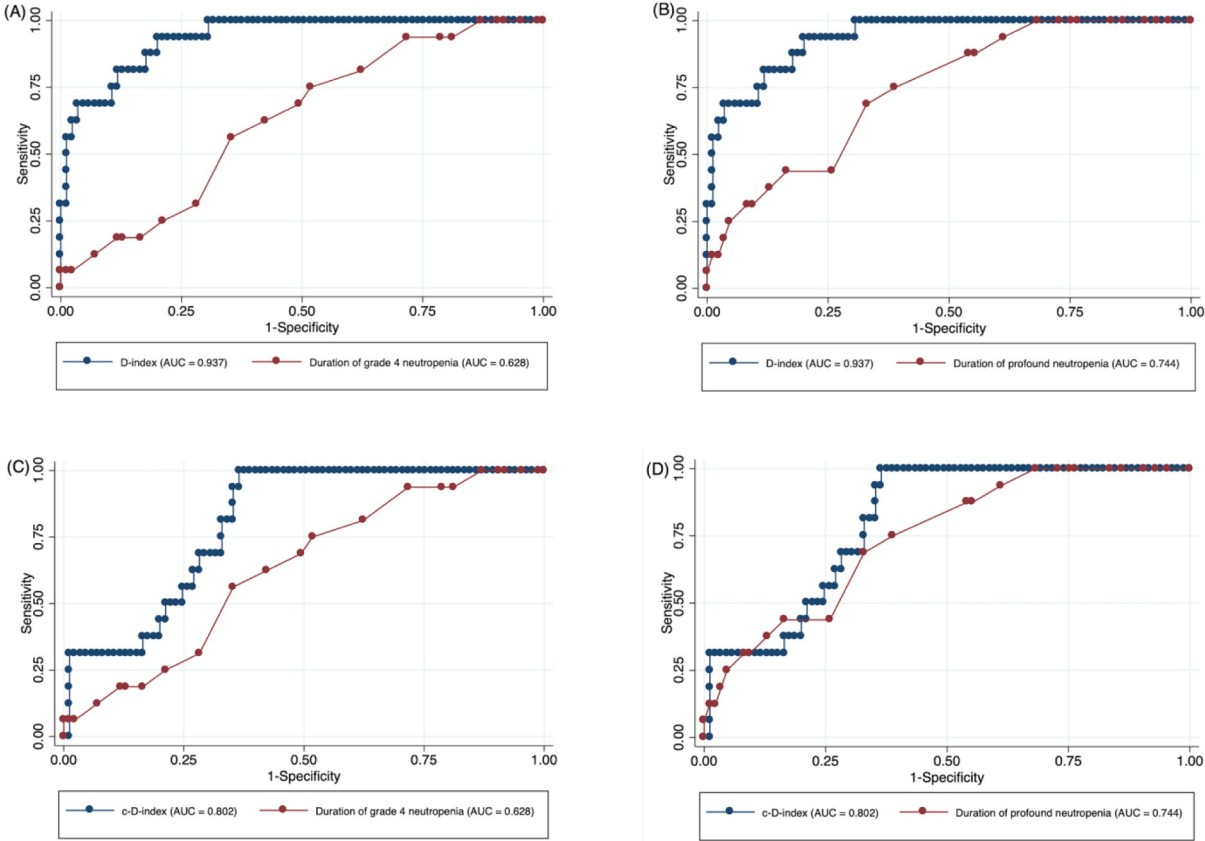

**Fig 3. ROC curve analysis between the D-index, c-D-index, duration of grade 4 neutropenia and profound neutropenia for predicting IFIs. A:** Comparison between D-index and grade 4 neutropenia (AuROC of 0.937 vs. AuROC of 0.628; $P < 0.001$), **B:** Comparison between D-index and profound neutropenia (AuROC of 0.937 vs. AuROC of 0.744; $P < 0.001$), **C:** Comparison between c-D-index and grade 4 neutropenia (AuROC of 0.802 vs. AuROC of 0.628; $P = 0.012$), **D:** Comparison between c-D-index and profound neutropenia (AuROC of 0.802 vs. AuROC of 0.744; $P = 0.295$).

particular cutoff. Moreover, the c-D-index could be used with other parameters to enhance the accuracy of diagnostic-driven therapy for IFIs while ANC was not fully recovered.

There was growing evidence of D-index implementation combined with serial serum galactomannan and imaging for diagnostic-driven antifungal therapy in acute leukemia patients [19]. Impacts of D-index on IFIs of pediatric AML [20] and different chemotherapy regimens for AML [21] were also reported. However, the different cutoffs on D-index and c-D-index in predicting IFIs in AML patients reflected the difference in populations among the studies. The D-index value of 6,200 and 3,875 was the optimal cutoff in the previous reports, while the optimal cutoff of 5,800 and 4,225 was addressed on the c-D-index. However, all the previous studies similarly supported the high NPV of the D-index and c-D-index for IFIs exclusion [8, 9]. Previous studies included all AML patients who received both first induction chemotherapy and salvage treatment for relapsed or refractory disease, whereas this study included only first-time receivers of systemic chemotherapy for induction treatment. Moreover, this study solely selected the patients with the first episode of febrile neutropenia to determine the utility of D-index in patients with febrile neutropenia that reflected the evaluation of complications during the sensitive period of induction remission.

The use of D-index in the context of other neutropenic complications was also reported. In AML patients with febrile neutropenia who received both induction and consolidation phases of chemotherapy, the c-D-index was initially found to be an independent risk factor for prolonged febrile neutropenia [22]. Subsequently, the study from the same group revealed that the c-D-index on Day 11 at 710 or above was associated with a higher cumulative incidence of febrile neutropenia than those with a c-D-index less than 710 (80% vs. 39%) [11]. Focusing on the pulmonary infection, the D-index of AML patients who received chemotherapy at the consolidation phase showed no difference between patients with and without any pulmonary infections. However, the sum of D-index from the induction to the consolidation phase was higher in the pulmonary infection group, suggesting the cumulative effect of neutropenia on infectious complications [23]. Additionally, there were studies on patients who underwent allogeneic stem cell transplantation (AlloSCT) and implemented D-index in predicting IFIs. The prevalence of IFIs in the studies of AlloSCT patients was 2–5%, with a high cutoff value of c-D-index at 10,644, showing 97.9% of NPV for IFIs [12]. Nevertheless, another study showed no difference in D-index value between those with and without IFIs, and the neutropenic duration was still a better marker for IFIs among AlloSCT recipients during the first 30 to 100 days after transplantation [24]. Interestingly, a randomized study on the patients with febrile neutropenia who received chemotherapy or AlloSCT comparing the use of c-D-index-guided early antifungal therapy (DET) with the cutoff of 5,500 or above versus empirical antifungal therapy (EAT) showed significantly reduced in antifungal usage in DET (32.5%) compared to EAT (60.2%) without survival difference. However, the IFIs incidence in DET and EAT groups were 0.5% and 2.5%, which was less than real-life studies that reported around 10–15% [10].

There were several limitations of this study to be addressed. Firstly, the study's retrospective design resulted in an inevitable loss of clinical information on IFIs manifestation, serial blood count during admission, and other significant data that might affect the study outcomes. The second limitation was the c-D-index cutoff was derived from the consideration of the first clinical manifestation of IFIs that may be varied among the physician's judgment to be noted in the medical record; however, there were confirmation tests included serum biomarker, tissue biopsy, and imaging to support in all the IFIs in this study. Moreover, the revised EORTC/MSG definition of IFIs using the threshold of serum galactomannan (GM) at 1.0 was changed from 0.5 in the previous version, reflecting an increased number of possible IPA cases in this study [16]. Furthermore, all AML patients in our center received mold-active antifungal prophylaxis with itraconazole, which may decrease the sensitivity of serum GM and underestimate the IFIs. Next, this study was entirely focused on AML patients who received induction remission and developed febrile neutropenia, resulting in limited in the generalizability. The different chemotherapy regimens in diverse disease populations could result in a different cutoff of the D-index in predicting IFIs or other infectious complications that needed further research. Lastly, the D-index and c-D-index calculation on the spreadsheet was relatively complicated in real-time clinical practice. The in-hospital pragmatic tool using the computerized program or applications for measuring area over the neutrophil curve will support their feasibility in real-life clinical care in patients who are estimated to develop iatrogenic neutropenia from a particular treatment. Using D-index surveillance in the anticipated neutropenia patients could subsequently show the benefit of early investigation and appropriate treatment of IFIs.

## Conclusions

In conclusion, the neutropenic burden measurement using the D-index in AML patients who developed the first episode of febrile neutropenia showed good performance in predicting IFIs. The D-index-guided strategy reduced the unnecessary use of antifungal agents and

potentially be used as an alternative decision tool before starting the empirical antifungal therapy for IFIs.

## Supporting information

**S1 File. The D-index calculation spreadsheet.**
(XLSX)

**S1 Checklist. STROBE statement—Checklist of items that should be included in reports of observational studies.**
(DOCX)

## Acknowledgments

We highly appreciate Mr. Nut Ruamrungsee, who orginally developed the spreadsheet for D-index calculation that was used in this study.

## Author Contributions

**Conceptualization:** Thanawat Rattanathammethee.

**Data curation:** Thanawat Rattanathammethee, Kawin Munsamai.

**Formal analysis:** Thanawat Rattanathammethee, Kawin Munsamai.

**Investigation:** Thanawat Rattanathammethee, Kawin Munsamai.

**Methodology:** Thanawat Rattanathammethee.

**Writing – original draft:** Thanawat Rattanathammethee.

**Writing – review & editing:** Thanawat Rattanathammethee, Kawin Munsamai, Teerachat Punnachet, Nonthakorn Hantrakun, Pokpong Piriyakhuntorn, Sasinee Hantrakool, Chatree Chai-Adisaksopha, Ekarat Rattarittamrong, Adisak Tantiworawit, Lalita Norasetthada.

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
