## [Decision Letter · Decision Letter 0]

17 Apr 2023

PONE-D-23-07952D-Index and invasive fungal infections (IFIs) in adult acute myeloid leukemia (AML) patients with the first episode of febrile neutropeniaPLOS ONE

Dear Dr. Rattanathammethee,

Thank you for submitting your manuscript to PLOS ONE. After careful consideration, we feel that it has merit but does not fully meet PLOS ONE’s publication criteria as it currently stands. Therefore, we invite you to submit a revised version of the manuscript that addresses the points raised during the review process.

We look forward to receiving your revised manuscript.

Kind regards,

Felix Bongomin, MB ChB, MSc, MMed, FECMM

Academic Editor

PLOS ONE

Journal Requirements:

Reviewers' comments:

Reviewer's Responses to Questions

**Comments to the Author**

1. Is the manuscript technically sound, and do the data support the conclusions?

Reviewer #1: Yes

Reviewer #2: Yes

2. Has the statistical analysis been performed appropriately and rigorously? 

Reviewer #1: Yes

Reviewer #2: I Don't Know

3. Have the authors made all data underlying the findings in their manuscript fully available?

Reviewer #1: Yes

Reviewer #2: Yes

4. Is the manuscript presented in an intelligible fashion and written in standard English?

Reviewer #1: Yes

Reviewer #2: Yes

5. Review Comments to the Author

Reviewer #1: Dr. Rattanathammethee et al. evaluated the relationship between area-based neutropenia index (D-index) and invasive fungal disease (IFD) in acute myeloid leukemia (AML) patients undergoing induction chemotherapy. Not only the D-index, but also the c-D-index in patients who developed IFD were significantly higher than the D-index in patients without IFD. The appropriate cut-off value of the c-D-index was set at 5625 with receiver-operating-curve (ROC) analysis, which was comparable to the previous study [Ref 8: J Clin Oncol 2009, 27: 3849-3854]. Using this cut-off value of the c-D-index and the prevalence of IFD to be 15.8%, the sensitivity, specificity, positive predictive value (PPV) and negative predictive value (NPV) of the c-D-index for the diagnosis of IFD were 68.8%, 68.2%, 28.9%, and 92.1%, respectively. Superiority of the D-index for the prediction of the development of IFD to simple duration of neutropenia seems to be more clearly revealed in this study compared to the original D-index study [Ref 8: J Clin Oncol 2009, 27: 3849-3854]. This is well written manuscript, and the results are interesting and exciting in my mind. I have only a few minor comments regarding Table 1.

1) Table 1.　Please define the comorbidities in this study

2) Table 1.　AML and IFI should be also spelled out in the Table 1.

3) Table 1.　What does “WBC per week” mean ? In addition, Pease clarify its unit.

Reviewer #2: The study highlighted neutropenic burden measuring using the D-index in AML patients who experienced their first episode of febrile neutropenia performed well in terms of predicting IFIs. The D-index-guided technique minimized the usage of superfluous antifungal medications and may be utilized as a substitute decision-making tool prior to the commencement of empirical antifungal therapy for IFIs.

To Facilite the use of D-index and c-D-index please include the calculation spreadsheet in the supplementary material

6. PLOS authors have the option to publish the peer review history of their article (what does this mean?). If published, this will include your full peer review and any attached files.

Reviewer #1: No

Reviewer #2: No

---

## [Author Response · Author response to Decision Letter 0]

17 Apr 2023

Reviewer #1

1) Table 1. Please define the comorbidities in this study

Response: I have added the details of comorbidities for both groups in the footnote of table 1, as suggested by the reviewer.

2) Table 1. AML and IFI should be also spelled out in the Table 1.

Response: I have already spelled out AML as "acute myeloid leukemia" and IFIs as "invasive fungal infections" in the footnote of table 1, as suggested by the reviewer.

3) Table 1. What does "WBC per week" mean? In addition, please clarify its unit.

Response: "WBC per week" referred to the WBC "count" per week, as specified in table 1. This phrase represented the frequency of serial blood cells counts per week. To provide a clearer and concise description, I have followed the reviewer's suggestion and added the unit of "time" after the term "WBC count per week".

Reviewer #2

To facilitate the use of D-index and c-D-index please include the calculation spreadsheet in the supplementary material.

Response: As suggested by the reviewer, I have provided the spreadsheet containing the calculation of the D-index in the supplement material or supporting information section.

---

## [Decision Letter · Decision Letter 1]

9 May 2023

D-Index and invasive fungal infections (IFIs) in adult acute myeloid leukemia (AML) patients with the first episode of febrile neutropenia

PONE-D-23-07952R1

Dear Dr. Rattanathammethee,

We’re pleased to inform you that your manuscript has been judged scientifically suitable for publication and will be formally accepted for publication once it meets all outstanding technical requirements.

Kind regards,

Felix Bongomin, MB ChB, MSc, MMed, FECMM

Academic Editor

PLOS ONE

Additional Editor Comments (optional):

Reviewers' comments:

Reviewer's Responses to Questions

**Comments to the Author**

1. If the authors have adequately addressed your comments raised in a previous round of review and you feel that this manuscript is now acceptable for publication, you may indicate that here to bypass the “Comments to the Author” section, enter your conflict of interest statement in the “Confidential to Editor” section, and submit your "Accept" recommendation.

Reviewer #1: All comments have been addressed

2. Is the manuscript technically sound, and do the data support the conclusions?

Reviewer #1: (No Response)

3. Has the statistical analysis been performed appropriately and rigorously? 

Reviewer #1: (No Response)

4. Have the authors made all data underlying the findings in their manuscript fully available?

Reviewer #1: (No Response)

5. Is the manuscript presented in an intelligible fashion and written in standard English?

Reviewer #1: (No Response)

6. Review Comments to the Author

Reviewer #1: (No Response)

7. PLOS authors have the option to publish the peer review history of their article (what does this mean?). If published, this will include your full peer review and any attached files.

Reviewer #1: No

---

## [Editor Report · Acceptance letter]

12 May 2023

PONE-D-23-07952R1 

D-Index and invasive fungal infections (IFIs) in adult acute myeloid leukemia (AML) patients with the first episode of febrile neutropenia 

Dear Dr. Rattanathammethee:

I'm pleased to inform you that your manuscript has been deemed suitable for publication in PLOS ONE. Congratulations! Your manuscript is now with our production department. 

Kind regards, 

on behalf of

Dr. Felix Bongomin 

Academic Editor

PLOS ONE